# Effect of Different Isometric Exercise Modalities on Myocardial Work in Trained Hypertensive Patients with Ischemic Heart Disease: A Randomized Pilot Study

**DOI:** 10.3390/jfmk10020108

**Published:** 2025-03-27

**Authors:** Giuseppe Caminiti, Giuseppe Marazzi, Maurizio Volterrani, Valentino D’Antoni, Simona Fecondo, Sara Vadalà, Barbara Sposato, Domenico Mario Giamundo, Matteo Vitarelli, Valentina Morsella, Ferdinando Iellamo, Vincenzo Manzi, Marco Alfonso Perrone

**Affiliations:** 1Department of Human Science and Promotion of Quality of Life, San Raffaele Open University, 00166 Rome, Italy; maurizio.volterrani@uniroma5.it (M.V.); matteo.vitarelli@uniroma5.it (M.V.); 2Cardiology Rehabilitation Unit, IRCCS San Raffaele, 00166 Rome, Italy; giuseppe.marazzi@sanraffaele.it (G.M.); valentino.dantoni@sanraffaele.it (V.D.); fecondosimona@outlook.it (S.F.); vadasara21@gmail.com (S.V.); barbara.sposato@sanraffaele.it (B.S.); valentina.morsella@sanraffaele.it (V.M.); 3Department of Systems Medicine, Tor Vergata University, 00133 Rome, Italy; jamundus20@libero.it; 4Division of Cardiology and Sports Medicine, Department of Clinical Sciences and Translational Medicine, University of Rome Tor Vergata, 0133 Rome, Italy; iellamo@uniroma2.it (F.I.); marco.perrone@uniroma2.it (M.A.P.); 5Department of Wellbeing, Nutrition and Sport, Pegaso Open University, 80143 Naples, Italy; vincenzo.manzi@unipegaso.it

**Keywords:** isometric exercise, hypertension, handgrip, ischemic heart disease, myocardial work

## Abstract

**Background:** Isometric exercise effectively reduces blood pressure (BP) but its effects on myocardial work have been poorly studied. For the present study, we compared acute changes in myocardial work during two different isometric exercises, namely, bilateral knee extension and handgrip, in patients with hypertension and underlying ischemic heart disease (IHD). Methods: This was a randomized pilot study in which 48 stable, trained patients with hypertension and IHD were enrolled. Patients were randomly assigned to perform a single session of bilateral knee extension (IKE) or handgrip (IHG) exercises or no exercise (control), with a 1:1:1 ratio. Both exercises were performed at 30% of maximal voluntary contraction and lasted three minutes. Echocardiography and BP measurements were performed at rest, during the exercise, and after ten minutes of recovery. **Results:** Both exercises were tolerated well, and no side effects occurred. During the exercise, the systolic BP increased significantly in the IKE group compared with the IHG and control groups (ANOVA *p* < 0.001). Left ventricular global longitudinal strain decreased significantly in the IKE group (−21%) compared with the IHG and control groups (ANOVA *p* 0.002). The global work index increased significantly in the IKE group (+28%) compared with the IHG and control groups (ANOVA *p* 0.034). Global constructive work and wasted work increased significantly in the IKE group compared with the IHG and control groups (ANOVA *p* 0.009 and <0.001, respectively). Global work efficiency decreased significantly in the IKE group (−8%) while remaining unchanged in the IHG and control groups (ANOVA *p* 0.002). **Conclusions:** Myocardial work efficiency was impaired during isometric bilateral knee extension but not during handgrip, which evoked a limited hemodynamic response.

## 1. Introduction

Exercise training is an effective non-pharmacological intervention for the prevention and treatment of hypertension [1]. Performing daily physical exercise has been recommended with a level of evidence of 1A by European and American guidelines on hypertension [2]. Among the different exercise modalities, isometric exercise (IE) has attracted the attention of researchers since it has shown potential advantages beyond its effectiveness in reducing blood pressure (BP) in normotensive and hypertensive subjects [3]. In particular, IE presents a convenient time-efficiency profile since short bouts of IE, lasting 11–20 min, have been shown to be as effective as longer sessions of aerobic or high-intensity interval training in reducing BP [4,5]. Furthermore, some types of isometric exercises, such as wall squats or handgrip exercises, can comfortably be performed at home without expense or with a very low financial burden for the patient [6]. Therefore, the use of IE appears to be particularly attractive in the long-term management of hypertensive patients, either alone or in addition to their anti-hypertensive drug therapy. Reducing BP is one of the mechanisms through which exercise, as part of cardiac rehabilitation programs, contributes to reducing cardiovascular risk in patients with already established ischemic heart disease (IHD) [7]. Regarding the use of IE in such complex and high-risk patients, an important aspect to take into consideration is the hemodynamic tolerability of this type of exercise, since an excessive increase in afterload during the exercise can induce unfavorable changes in left ventricular (LV) filling pressure, with a consequent increase in myocardial wall tension and myocardial oxygen consumption [8]. Therefore, a preliminary assessment of the hemodynamic response to IE could help to identify the most well-tolerated IE modality and intensity. While, in the past, the measurement of hemodynamic parameters was limited to an experimental setting because of the need to use invasive procedures, the utilization of speckle tracking echocardiography permits today’s clinicians to considerably expand hemodynamic assessment into clinical practice. This ultrasound technique, through the measurement of strain and strain rate, assesses the percentage of deformation of the myocardium and allows clinicians to perform a comprehensive evaluation of LV and left atrial (LA) function [9,10,11]. Despite several factors, including image quality, the modest reproducibility of measurements, and cardiac rhythm, being able to affect the accuracy and reliability of strain-dependent measurements such as LV global longitudinal strain (LVGLS) [12], speckle tracking echocardiography has demonstrated significant value for patient management and in guiding clinicians toward optimum treatment strategies, particularly in patients with advanced cardiac disease [13,14]. Moreover, it allows them to assess the changes induced by exercise in different scenarios [15,16,17,18]. In addition, the assessment of myocardial work makes it possible to reconstruct, in a non-invasive way, the LV pressure–volume loops [19], giving the opportunity to study relationships among preload, afterload, and myocardial contractility [20]. The purpose of the present study was to compare the acute changes in myocardial work and LA function occurring during two different isometric exercise modalities, namely, isometric knee extension and handgrip exercises, in hypertensive patients with IHD.

## 2. Materials and Methods

### 2.1. Population

A total of 48 patients (both male and female) participating in secondary prevention and rehabilitation programs at San Raffaele IRCCS in Rome were enrolled. We used the following inclusion criteria: age over 50 years old; an established diagnosis of hypertension; a previous diagnosis of IHD, for which the following diagnostic criteria of IHD were adopted: previous acute coronary syndrome, including ST-elevation myocardial infarction (STEMI), non-ST-elevation myocardial infarction (NSTEMI) and unstable angina; previous percutaneous coronary intervention (PCI) or coronary artery bypass grafting (CABG), or both; stable clinical conditions: patients must not have been hospitalized in the previous six months and their pharmacological therapy must have remained unchanged for at least three months before the enrolment; being physically active: we enrolled patients who declared that they performed moderate-intensity aerobic exercise for at least 150 min/week (walking, cycling, or swimming) [21]. We adopted the following exclusion criteria: signs and/or symptoms of myocardial ischemia or threatening arrhythmias during the resting assessment or during the ergometric test; incomplete revascularization; permanent atrial fibrillation or a history of recurrent episodes of atrial fibrillation; and baseline BP levels at rest over 160/100 mmHg, despite the current drugs therapy. Subjects with severe heart valve diseases, with a diagnosis of hypertrophic cardiomyopathy, with a previous diagnosis of chronic heart failure, or with signs or symptoms of heart failure during the screening visit were also excluded. The following extracardiac conditions were considered among the exclusion criteria: low levels of hemoglobin (below 10.5 g/dL); a diagnosis of advanced chronic pulmonary disease (GOLD stage III–IV); a diagnosis of symptomatic peripheral artery disease (Leriche–Fontaine stage II–IV). The study complied with the Declaration of Helsinki and was approved by the local Ethics Committee of San Raffaele IRCCS (protocol number 26/2023). All patients gave written informed consent before entering the study.

### 2.2. Study Design

The study design, which is summarized in Figure 1, complies with CONSORT guidelines [22]. This research was conceived as a three-armed parallel randomized pilot study in which patients were alternately allocated to one of the following groups, at a ratio of 1:1:1, according to exercise: (1) isometric knee extension (IKE); (2) isometric handgrip (IHG); (3) control, no exercise. The randomization code was developed by a computer random-number generator to select random permuted blocks. This was a single-blinded study in which the operator who performed the analysis of the echocardiography variables was unaware of the patient allocation. The recruitment period started in September 2023 and was concluded in February 2024. All patients were evaluated during a preliminary screening that included the following examinations: clinical history collection; measurement of anthropometric parameters, including the resting heart rate (HR), resting systolic and diastolic BP; and carrying out a symptom-limited ergometric test. Those patients who fit the inclusion/exclusion criteria were asked to participate in the study. The patients were then summoned for a trial session, during which they were encouraged to try out the devices and exercises that were the object of the study in order to familiarize themselves with the experimental protocol and with the devices’ use. For each patient, the experimental session was performed within 10 days of the screening visit.

### 2.3. Echocardiography

Transthoracic echocardiography: In the echocardiographic examinations, the Vivid E95^®^ cardiovascular ultrasound device (GE Healthcare, Chicago, IL, USA) with a 4.0 MHz transducer was used for the entire duration of the study. Each examination was performed with one-lead electrocardiographic monitoring, and the imaging windows with the respective measurements were obtained according to the current guidelines of the European Association of Cardiovascular Imaging [23]. All acquired echocardiographic images were digitally archived and their analysis was performed offline. During the review process, an experienced technician performed strain measurements using proprietary software (version 10.8, EchoPAC; GE Vingmed Ultrasound, Horten, Norway). The technician performing the offline analyses was unaware of each patient’s assignment group. Left ventricular diastolic function was assessed using the E/A ratio, defined as the ratio of the E-wave (corresponding to the peak left ventricular filling velocity in early diastole) to the A-wave (corresponding to the peak flow velocity in late diastole). Color tissue Doppler tracings were performed in the four-chamber view, and the range gate was positioned at the lateral mitral annular segments. The E/e’ ratio was defined as the ratio of the E-wave velocity to the average of the septal and lateral e’-wave velocities of the left ventricle. The volume of the left atrium (LA) was obtained from apical four-chamber and two-chamber views at the end of systole, prior to the adoption of Simpson’s biplanar disc method and prior to the opening of the mitral valve. The LA volume index (LAVI) was obtained by dividing the LA volume by the body surface area of the study subjects. Measurements of the left ventricular end-diastolic volume (LVEDV) and left ventricular end-systolic volume (LVESV) were calculated from the apical windows of two and four chambers; subsequently, the LVEF was calculated using Simpson’s modified method. Stroke volume (SV) was then calculated as EDV−ESV, while cardiac output (CO) was calculated as HR × SV, and ejection fraction (EF) as EF = (EDV − ESV)/EDV. Measurements of GLS were obtained from four-chamber, three-chamber, and two-chamber views. The software measured the maximum negative value of the deformation during systole and this value was considered to be the maximum contractility for each segment. The software is able to automatically detect the endocardial LV boundaries; however, whenever deemed appropriate, the images can be modified to conform with the displayed LV boundaries. Therefore, LVGLS was calculated by considering the mean values of each segment. LA deformation was assessed using the four-chamber and two-chamber views. The software automatically plotted the endocardial and epicardial contours of the LA, using R-R gating in which the R-wave represented the starting point. Again, manual adjustments were made if necessary. A series of control points was automatically placed on the central curve of the myocardial wall in the reference phase, based on the endocardial and epicardial contours that were plotted. The software program generated longitudinal deformation curves for each segment and calculated the average curve for each segment. LA reservoir strain, conduit strain, and contractile strain were obtained by splitting the longitudinal strain measurements. The myocardial work was assessed from the closure to the opening of the mitral valve. PACS was defined as a positive peak during the onset of left ventricular diastole, prior to the onset of the atrial systolic phase, whereas PALS was defined as a positive peak during left ventricular systole, at the end of the atrial diastolic phase. A 17-segment bull’s eye was obtained with the segmental and global work index (GWI), corresponding to the area within the total work curve from mitral valve closure to its opening. Global constructive work (GCW) was defined as the work performed during shortening in systole and the negative work performed during lengthening in isovolumetric relaxation. Global wasted work (GWW) corresponds to the work performed during shortening in isovolumetric relaxation, plus the negative work performed during lengthening in systole. Global work efficiency (GWE) was defined as constructive work, divided by the sum of constructive work and wasted work. Using pulsed-wave Doppler recordings at mitral and aortic valve levels, the timing of the valve events was identified. Confirmation of the valve events was performed by a 2D evaluation of the long axis and apical view [24].

### 2.4. Experimental Sessions

All experimental sessions were performed during the morning, between 9:00 and 11:30 am. Patients were asked to avoid drinking coffee and alcohol for at least 24 h before the session. A light breakfast was permitted at least two hours before the experimental session. Bilateral knee extension group: the experiments were conducted on a knee flex/extension dynamometer (Technogym Wellness System, Technogym, Cesena, Italy). Patients were seated on the dynamometer, with their backs reclined at 120 degrees from the horizontal plane and with their knees bent at 90 degrees from the trunk. The seat was individually regulated so that the axis of rotation around the dynamometer shaft was adjacent to the lateral femoral condyle of each subject’s right leg. Patients positioned their legs under the knee extension/flexion attachment arm of the dynamometer. Both of their arms were positioned along the trunk. The sonographer responsible for the acquisition of echocardiographic data was positioned on the left side of each patient. Patients had a manual sphygmomanometer cuff placed on their right arm. Handgrip exercise group: experiments were conducted with patients lying on a bench, with their backs reclined at 120 degrees from the horizontal plane. The sonographer was located on the patient’s left side and a sphygmomanometer cuff was placed on the arm contralateral to the dominant one that was used for the handgrip exercise (GIMA, S.P.A., Gessate (MI), Italy). For both bilateral knee extension and handgrip exercises, the determination of the maximal voluntary contraction (MVC) consisted of three maximal contractions, each one lasting 3–5 s, with 1 min of rest between contractions. For both exercises, the intensity was set at 30% of the patients’ MVC. The exercise phase lasted 3 min, and, during this time, the patients had to exert constant force. During the exercise, patients were instructed to breathe at a normal rhythm and depth in order to avoid Valsalva maneuvers. The echocardiography examination started after the first minute of exercise. Control group: patients were positioned lying on a bench, with their backs reclined at 120 degrees from the horizontal plane. The sonographer was positioned on the left side of the patient and a sphygmomanometer cuff was placed on the patient’s right arm. For patients in the IKE and IHG groups, echocardiography acquisitions and BP measurements were made: (1) at rest; (2) during the exercise phase, starting one minute from the beginning of the isometric exercise; (3) ten minutes after the end of the isometric exercise. For patients in the control group, echocardiographic acquisitions and BP measurements were made using the same timing as for the other two groups, but the patients of this group remained at rest during the entire procedure.

### 2.5. Statistical Analysis

This research was conceived as a pilot study; no formal a priori power analysis was performed. Therefore, the sample size was determined by feasibility and patients’ availability [25]. Data were expressed as the mean ± SD. The assumption of normality was checked using the Shapiro–Wilk hypothesis test. Comparisons of the changes occurring in different variables during the three different stages of the experimental session, at rest, during exercise, and at recovery, were compared using a repeated-measures two-way ANOVA with Bonferroni corrections for post hoc testing. The level of significance was set at *p* < 0.05. Categorical variables were expressed as absolute and percentage values and were compared with the chi-square test. A statistical program, IBM SPSS Statistics v26.0, was used for the processing, presentation, and statistical analysis of the data.

## 3. Results

From the initial number of 69 patients recruited, 19 declined to participate in the study. Forty-eight patients were then randomized, and each group included sixteen patients. All patients included in the study completed the protocol and their data were analyzed. The baseline anthropometric, clinical, and echocardiography data are reported in Table 1. At baseline, there were no significant differences in the clinical characteristics between the three study groups. Out of the 48 participants, 36 (75%) had a previous STEMI. Of these 36 patients, 23 (64%) had an anterior STEMI and 13 (36%) had an inferior STEMI. LVEF ranged between 38% and 57%. Out of the 48 participants, 19 (39%) had LVEF values between 40% and 50% and 3 (6%) had LVEF values below 40%. Finally, out of the 48 participants, 37 (77%) had a diagnosis of multivessel coronary disease. All patients were treated with anti-platelets agents, statins and beta-blockers. Overall, they were taking on average 2.4 ± 1.3 anti-hypertensive drugs. There were no differences in pharmacological therapy between the three groups. Both exercise sessions were tolerated well, and no side effects occurred.

### 3.1. Intra-Group Changes

During the exercise phase, compared with baseline values, systolic BP significantly increased in the IKE group (+36.0 ± 1.7 mmHg) while it remained unchanged in the IHG (+3.5 ± 1.7mmHg) and control groups (−2.5 ± 0.8 mmHg). LVGLS decreased significantly in the IKE group (−21%) while remaining unchanged in the IHG (−3%) and control groups (+2%). The IKE group presented a significant increase in GWI (+28%, *p* 0.003), while no significant changes in GWI occurred in the IHG and control groups. GCW and GWW increased significantly in the IKE group (+30% and +56%, respectively); no significant changes in GCW and GWW were observed in the IHG group (+4% and +11%, respectively) and control group (+2% and −3%, respectively). GWE decreased −7%, significantly, in the IKE group and remained unchanged in the IHG group (−0.4%) and control group (+1%). PALS was unchanged in the IKE (−4%), IHG (−3%), and control (−1%) groups. No significant changes in diastolic BP, E/e’, PALS, LAVI, LVEF, SV, and CO occurred during exercise in comparison to the baseline values in the two active groups and control group. During the recovery phase, compared with baseline values, systolic BP decreased significantly in the IKE group compared with the IHG and control groups. GWW decreased significantly in the IKE and IHG groups and was unchanged in the control group.

### 3.2. Between-Groups Changes

The increase in systolic BP observed in the IKE group was significantly greater than in the IHG and control groups (ANOVA: F = 7.13; *p* < 0.001) (Figure 2). The reduction in GLS in the IKE group was significantly greater than in the IHG and control groups (ANOVA: F = 3.8; *p* 0.002). The increase in GWI (ANOVA: F = 2.7; *p* 0.034), GCW (ANOVA: F = 3.81; *p* 0.009), and GWW (ANOVA: F = 6.20. *p* < 0.001) in the IKE group was significantly greater compared with the IHG and control groups (Figure 3 and Figure 4). The IKE group presented a significant decrease in GWE compared with the IHG and control groups (Figure 5). No between-group changes occurred regarding diastolic BP, E/e’, PALS, LVEF, SV, and CO. During the recovery phase, the decrease in systolic-BP in the IKE group was significantly greater than in the IHG and control groups (ANOVA: F = 2.4. *p* 0.032) (Figure 2).

## 4. Discussion

The utilization of IE in the context of the rehabilitation of patients with IHD has, thus far, been prevented by the concern that its use may result in an excessive increase in myocardial oxygen consumption. However, it remains an attractive exercise modality in the long-term management of hypertension in these patients since, besides its anti-hypertensive efficacy, it presents a valuable time-efficiency and cost-sparing profile [6]. Investigating the hemodynamic response to different types of isometric exercise can help to identify the best-tolerated modality and is useful when choosing the most appropriate intensity and duration of the exercise itself. In the present study, the hemodynamic response observed during IE was very different between bilateral knee extension and handgrip exercises. The IKE group presented a significant increase in systolic BP, which was coupled with a decrease in GLS and a rise in myocardial work, with a prevalence of GWW over GCW and a consequent reduction in GWE. These changes indicate, overall, a maladaptive LV contractile response during IKE. This kind of response is similar to that previously observed by our group in a similar population [26], and by Beaumont et al. [27] in healthy subjects performing bilateral knee extension, respectively, at 30% and 40% of MV. However, compared with our previous study [26], there are some differences that should be underlined: the rate of GWE loss during exercise in the present study appears to be lower when compared with that observed in the previous study (−7% vs. −18%); moreover, contrary to what happened in the previous research [26], changes in E/e’ and PALS in the IKE group during the exercise did not reach statistical significance. We believe that these results may depend on the difference in the fitness level of the patients enrolled in the two studies since the previous study enrolled only sedentary patients, while, in the present research, being physically active was one of the inclusion criteria. This hypothesis is supported by recent findings in the literature: a four-week exercise training regimen was effective in improving GWE in hypertensive patients [28]. In a study conducted on healthy subjects, Rovithis et al. [29] demonstrated that active men did not show a statistically significant change in the E/e’ ratio during isometric handgrip exercises; conversely, the inactive participants’ E/e’ ratio was higher during isometric handgrip exercises. Considering that both the E/e’ ratio and PALS are considered non-invasive metrics for LV filling pressure [30,31], we can hypothesize that IE performed with bilateral knee extension exercises had a lower impact on diastolic function in our patients because of their active physical status. However, we cannot rule out that these results could be affected by the small sample size, and they need to be confirmed in further and larger trials. Moreover, direct comparisons of the hemodynamic response to IE between physically active versus sedentary IHD patients are required. The IHG group showed only a slight increase in systolic BP, no changes in LV filling pressure metrics, and a mild, non-significant, increase in myocardial work that was balanced between GCW and GWW, so as not to observe changes in contractile efficiency. This response of the IHG group was statistically not different from that observed in the control group, indicating that isometric handgrip exercises performed at 30% of MVC had neutral effects on the central hemodynamic parameters. This result agrees with previous research investigating the BP response to isometric handgrip exercises; overall, this shows that when performed at low intensities, this exercise does not result in cardiovascular overload [32,33]. Since the timing of the exercise and the percentage of MVC utilized were the same between the IKE and IHG groups, our data suggest that the amount of muscle mass involved in the IE was the main determinant of the BP response during the exercise phase. This result appears also to be in line with the current literature [34]. Kounoupis et al. [35] showed that performing small-muscle mass, isometric, and dynamic resistance exercises evoked equal increases in BP. We observed a greater drop in systolic BP ten minutes after the end of the exercise in the IKE group compared with the IHG group. This result complies with previous reports showing that isometric exercise involving a larger muscle mass is associated with a significantly greater post-exercise hypotensive response in the first minutes of recovery, compared with a smaller muscle mass, as seen with those patients involved in the isometric handgrip training [36]. The effectiveness of isometric handgrip training in eliciting acute reductions in BP has recently been questioned by a meta-analysis demonstrating that while a single session of isometric handgrip exercises did not produce significant BP reductions, isometric handgrip training decreased SBP and DBP values by 6.7 mmHg and 4.5 mmHg, respectively [37]. Another aspect to consider is the complexity of pharmacological treatments already taken by the patients, which might have interfered with the expected BP reduction induced by IE. While a single bout of isometric handgrip exercise, performed at low intensity, was effective in lowering BP in healthy and pre-hypertensive individuals [38], the same exercise did not induce significant reductions in BP in patients with coronary artery disease [39]. Moreover, our results on post-exercise BP could be influenced by the fact that the results were obtained after single prolonged contractions (three minutes for both exercises); it is possible that shorter repeated contractions, as is generally the case during an exercise session, might have produced different results, as shown by Souza et al. [40] in elderly hypertensive patients. Considering its mild impact on systolic BP and its neutral effects on diastolic LV metrics and GWE, low-intensity isometric training using a handgrip appears to have a safer profile in patients with IHD. These considerations make this type of exercise a suitable non-pharmacological intervention for treating hypertension in IHD patients. However, the lack of significant BP reduction in the post-exercise phase that we observed in our study requires further research on this topic. Future studies should assess the safety and effectiveness of moderate and high-intensity isometric handgrip training in IHD patients and investigate whether the BP reductions achievable with moderate to high-intensity isometric handgrip training are comparable to those of bilateral isometric knee extension exercises. We think that this study adds useful information regarding the choice of exercise modality aimed at the non-pharmacological management of hypertension in patients with IHD and underlines the potential advantages of non-invasively assessing the hemodynamic response before starting an IE protocol. Limitations: since this research was conceived as a pilot study, it included a small sample size and lacked adequate power analysis. These considerations make it unclear whether the study was adequately powered to detect meaningful differences. Moreover, the lack of allocation concealment may have affected the study results. Therefore, caution is needed when drawing general conclusions from the data provided in this study. The study enrolled only three female patients in the active arms of the study cohort. Due to the unbalanced gender distribution of our sample, the results of this research cannot be extended to apply to women. Further studies focusing on the effects of isometric exercise in female patients with hypertension and IHD will, therefore, be necessary. Bilateral isometric knee extension and handgrip exercises were both performed at 30% of MVC; we cannot rule out that IE performed at different intensities could evoke a different myocardial response. In this study, LV hemodynamic changes were assessed noninvasively through speckle tracking echocardiography and, in particular, myocardial work. Although myocardial work is believed to provide a more accurate assessment of the LV function than LVEF and LVGLS, it has several technical limitations: these include poor image quality that does not allow the correction of endocardial border delineation and partial load-dependency [41]. Overall, these limitations could prevent the extension of the results of the present study. Finally, the use of myocardial work to investigate the cardiac response to exercise is, so far, very limited, and further research is needed to establish its utility in this field. It is worth mentioning that patients enrolled in the present study, who have chronic LV damage and, in several cases, mild LV dysfunction, had some features of heart failure and were at high risk of developing full heart failure syndrome. However, at the time of recruitment, none of the participants had signs or symptoms of heart failure (or had had them in the past) and were not taking the full drug treatment specifically used for heart failure. We considered those patients having IHD without heart failure. It is possible, however, that the response of myocardial work to IE was different across EF values. In this study, we did not consider this point, due to the small sample size, and this point should be investigated in further research. Despite there not being statistical differences in pharmacological therapy between the three groups, it is possible that different drugs may have affected the hemodynamic response to IE in a different way. However, this study, due to its design and the small sample size, is not suitable for providing this type of information and this point needs to be clarified in further investigations.

## 5. Conclusions

Our data suggests that in hypertensive patients with underlying IHD, performing IE with a handgrip seems a viable choice in terms of hemodynamic tolerability. Future studies should clarify whether handgrip IE is also effective in lowering BP in these patients.

## Figures and Tables

**Figure 1 jfmk-10-00108-f001:**
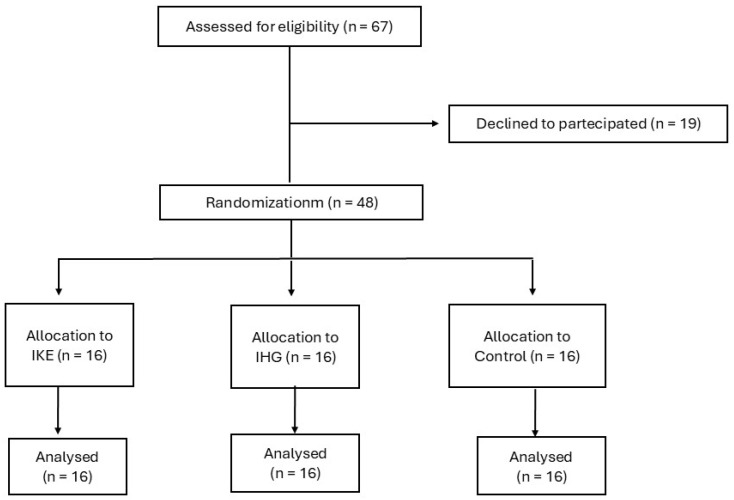
Study flow-chart.

**Figure 2 jfmk-10-00108-f002:**
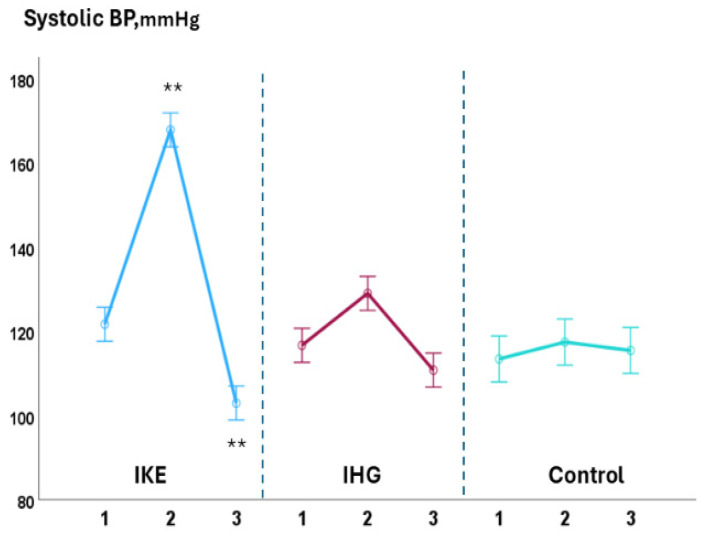
Changes in systolic BP during the three phases of exercise (1 = baseline; 2 = exercise; 3 = recovery) in the IKE, IHG, and control groups. A significant increase in systolic BP was observed during the exercise phase in the IKE group compared with the IHG and control groups. During the recovery phase, systolic BP significantly decreased in the IKE group compared with the IHG and control groups. ** = *p* < 0.05 versus the active group and control group.

**Figure 3 jfmk-10-00108-f003:**
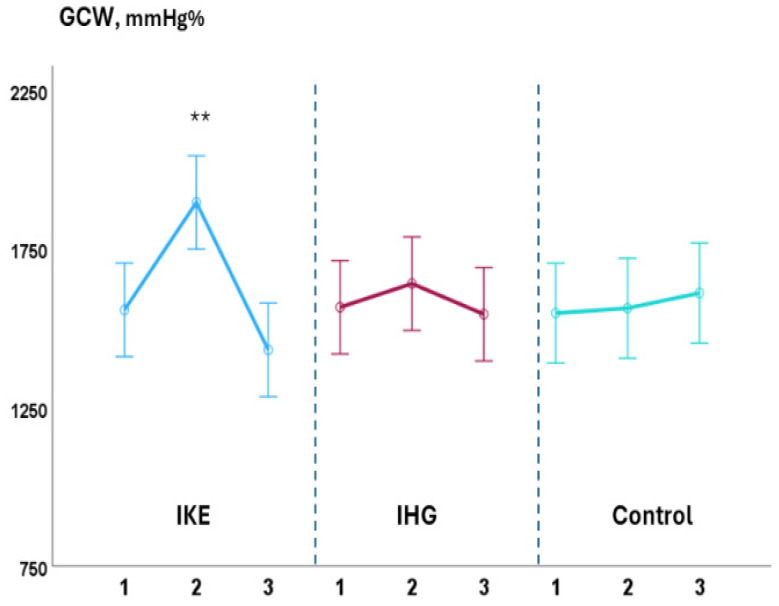
Changes in GCW during the three phases of exercise (1 = baseline; 2 = exercise; 3 = recovery) in the IKE, IHG, and control groups. A significant increase in GCW was observed during the exercise phase in the IKE group compared with the IHG and control groups. ** = *p* < 0.05 versus the active group and control group.

**Figure 4 jfmk-10-00108-f004:**
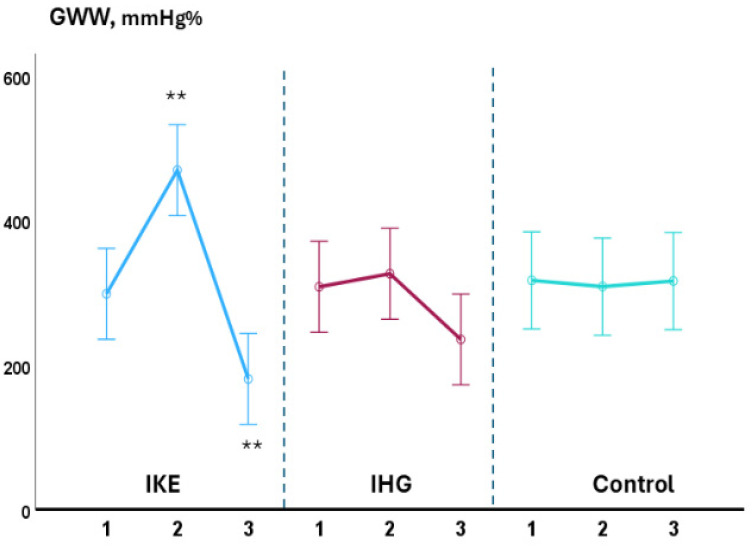
Changes in GWW during the three phases of exercise (1 = baseline; 2 = exercise; 3 = recovery) in the IKE, IHG, and control groups. A significant increase in GWW was observed during the exercise phase in the IKE group compared with the IHG and control groups. During the recovery phase, GWW values significantly dropped in the IKE group compared with the IHG and control groups. ** = *p* < 0.05 versus the active group and control group.

**Figure 5 jfmk-10-00108-f005:**
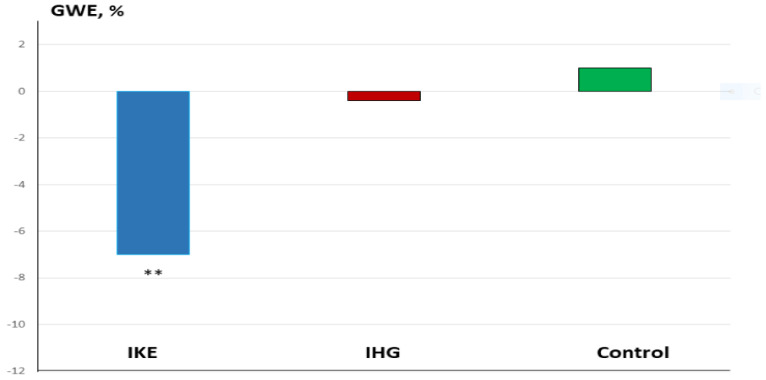
Percent changes in GW (exercise vs. baseline) in the IKE, IHG, and control groups.** = *p* < 0.05 versus the active group and control group.

**Table 1 jfmk-10-00108-t001:** Clinical and echocardiography parameters of patients enrolled in the study.

	**IKE** **(n = 16)**	**IHG** **(n = 16)**	**Control** **(n = 16)**
Age, years	64.9 ± 14.7	63.4 ± 16.1	64.2 ± 13.5
Male/female, n	14/2	14/1	7/1
BMI, kg/m^2^	27.1 ± 5.5	28.2 ± 8.0	27.5 ± 7.3
BSA, m^2^	1.88 ± 1.1	1.93 ± 0.8	1.91 ± 1.3
Previous STEMI	13 (81)	11 (69)	12 (75)
PCI, n (%)	11 (69)	10 (62)	10 (62)
CABG, n (%)	8 (50)	7 (44)	9 (56)
Physical activity (min/week)	174	178	173
HR, bpm	66.8 ± 13.6	63.5 ± 11.4	65.2 ± 8.2
SBP, mmHg	128.9 ± 27.5	135.2 ± 33.8	136.7 ± 32.1
DBP, mmHg	81.6 ± 9.2	80.9 ± 12.4	81.8 ± 1.3
Comorbidities			
Diabetes, n (%)	3 (19)	2 (12)	3 (19)
Hypercholesterolemia, n (%)	14 (87)	15 (94)	14 (87)
GFR < 60 mL/min/1.73 m^2^	3 (19)	4 (25)	2 (25)
Previous smoke habit, n (%)	8 (50)	10 (62)	8 (50)
Echocardiography			
LVEDV, mL	140.4 ± 33,4	136.3 ± 32.4	137 ± 32.4
LVESV, mL	70.5 ± 11.7	67.0 ± 26.3	69.4 ± 19.4
LVEF, %	50.3 ± 7.4	49.2 ± 8.2	50.3 ± 11.4
LVGLS, %	−12.4 ± 3.8	−13.1 ± 3.5	−12.9 ± 3.1
GWI, %	1150.5 ± 412.6	ì284.9 ± 492.0	1198 ± 331.6
GCW, %	1601.3 ± 491.8	1678 ± 558.3	1523 ± 341.9
GWW, %	338.7 ± 127.7	345.2 ± 198.9	344.3 ± 166.5
GWE, %	82.5 ± 11.7	82.8 ± 9.7	81.5 ± 13.1
DT, ms	208.7 ± 60.4	197.4 ± 37.5	211.7 ± 44.5
E, cm/s	48.1 ± 9.0	50.6 ± 13.0	52.3 ± 12.3
A, cm/s	66.4 ± 16.1	67.3 ± 13.9	66.9 ± 14.1
E/A ratio	0.75 ± 0.16	0.75 ± 0.22	0.78 ± 0.13
e’, cm/s	5.9 ± 1.5	6.2 ± 2.2	6.3 ± 2.2
E/e’ ratio	8.1 ± 1.9	8.1 ± 2.6	8.3 ± 2.3
TRV, m/s	1.9 ± 0.4	2.1 ± 0.1	1.7 ± 0.4
PALS, %	19.7 ± 8.5	21.4 ± 6.0	19.2 ± 6.0
PACS, %	−13.9 ± 3.5	−15.4 ± 3.5	−15 7 ± 4.3
LAVI, mL/m^2^	31.6 ± 7.2	34.2 ± 10.3	32.72 ± 9.1
SV, mL	70.3 ± 14.3	72.0 ± 14.4	69.3 ± 16.4
CO, L/min	4.7 ± 1.1	4.6 ± 1.0	4.5 ± 1.6
Treatment			
Anti-platelets agents, n (%)	16 (100)	16 (100)	16 (100)
ACE-Is/ARBs, n (%)	15 (94)	16 (100)	15 (94)
Beta-blockers, n (%)	16 (100)	16 (100)	8 (100)
Tiazidics, n (%)	4 (25)	5 (31)	5 (31)
CCAs, n (%)	5 (31)	6 (37)	7 (44)
Ranolazine, n (%)	3 (19)	2 (12)	2 (12)
Furosemide, n (%)	1 (6)	2 (12)	-
Statins, n (%)	16 (100)	16 (100)	16 (100)
Ezetimibe, n (%)	13 (81)	12 (75)	13 (81)

BMI = body mass index; BSA = body surface area; STEMI = ST-elevation myocardial infarction; PCI = percutaneous coronary intervention; CABG = coronary artery bypass grafting; HR = heart rate; SBP = systolic blood pressure; DBP = diastolic blood pressure; GWI = global work index; GCW = global constructive work; GWW = global waste work; GWE = global work efficiency; DT = deceleration time; TRV = tricuspid regurgitation velocity; PALS = peak atrial longitudinal strain; PACS = peak atrial contraction strain; SV = stroke volume; CO = cardiac output; ACE-Is = angiotensin-converting enzyme inhibitors; ARBs = angiotensin receptor blockers; CCAs = calcium-channel antagonists.

## Data Availability

The data presented in this study are available upon request from the corresponding authors.

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
