# Peer review of "Effect of Different Isometric Exercise Modalities on Myocardial Work in Trained Hypertensive Patients with Ischemic Heart Disease: A Randomized Pilot Study"

_jfmk, 2025, doi:10.3390/jfmk10020108_

Round 1

Reviewer 1 Report

Comments and Suggestions for Authors

General Assessment

The manuscript explores the acute effects of different isometric exercise modalities on myocardial work in trained hypertensive patients with ischemic heart disease (IHD). The study is well-structured, and the research question is relevant, given the increasing interest in non-pharmacological interventions for hypertension management. The methodology is appropriate, utilizing a randomized controlled trial design with echocardiographic assessments, including speckle tracking, to evaluate myocardial work.

However, several critical aspects require clarification and improvement before the manuscript can be considered for publication. Notably:

  • The study is not explicitly labeled as a randomized controlled trial (RCT), despite having characteristics of one.
  • The randomization process is unclear, and there is no mention of allocation concealment or blinding procedures.
  • The study does not conform to CONSORT guidelines, which are the standard for reporting RCTs.
  • Figure 1 (study flow diagram) does not align with CONSORT requirements, missing essential details such as participant recruitment, randomization method, and dropouts.
  • The title does not specify the study design, which is a requirement for transparency in clinical research.

The findings suggest that handgrip exercise may be a more hemodynamically tolerable modality compared to bilateral knee extension. While this conclusion is supported by the data, additional contextual discussion regarding its clinical significance and potential implications for exercise prescription in hypertensive patients with IHD would strengthen the manuscript.

I appreciate the opportunity to review this study, which addresses an important topic in cardiovascular rehabilitation and exercise science. The authors’ efforts in conducting this research are commendable, and I hope my comments contribute to enhancing the clarity and impact of their work.

Key Strengths

- Relevant Research Question: The study addresses a pertinent clinical issue regarding the safety and efficacy of isometric exercise in hypertensive patients with IHD.
- Methodological Rigor: The use of echocardiographic assessments, particularly speckle tracking, is a strength, providing objective and reliable data on myocardial work.
- Well-Organized Presentation: The manuscript follows a logical structure, and the results are presented with appropriate statistical analyses.

Major Concerns and Recommendations

  1. Study Design and Transparency Issues

Unclear Randomization and Lack of Explicit RCT Declaration

  • The study appears to be a randomized controlled trial (RCT), but the authors do not explicitly declare this in the manuscript.
  • There is no clear explanation of how the randomization was conducted. Was it computerized? Was there allocation concealment?
  • The absence of blinding details raises concerns about potential biases in data collection and interpretation.
  • The study should explicitly declare itself as an RCT in the title, abstract, and methods section.

Suggestion: Clearly state in the methods section that this is an RCT and provide details on:

  • Randomization method (e.g., computer-generated, sealed envelopes, etc.).
  • Allocation concealment (e.g., who assigned participants to groups, how was bias minimized?).
  • Blinding procedures, if any.

  1. Lack of Compliance with CONSORT Guidelines

No Mention of CONSORT, Missing Key Information

  • The manuscript does not cite the CONSORT guidelines, which are essential for reporting RCTs.
  • Essential components missing:
    • No flow diagram of participant allocation following CONSORT structure.
    • No reporting of dropouts, exclusions, or adherence rates.
    • No explicit statement of primary and secondary outcomes in line with CONSORT recommendations.

Suggestion:

  • The authors must cite CONSORT guidelines and restructure their methodology and results to align with them.
  • Include a proper CONSORT-compliant flow diagram (see next section).
  • Add transparency regarding participant losses, exclusions, and adherence to the intervention.

  1. Figure 1 (Study Flow Diagram) is Incomplete and Non-CONSORT Compliant

Critical Missing Information:

  • The current Figure 1 does not provide the required details for a clinical trial.
  • Missing elements:
    • Total number of participants screened, excluded, and randomized.
    • Justification for exclusions before randomization.
    • Breakdown of how many participants completed each stage of the study.
    • Dropouts and reasons for attrition in each group.

Suggestion

  • Redesign Figure 1 to follow CONSORT-compliant flow diagram standards, clearly showing:
    1️Enrollment → Number of patients assessed for eligibility.
    2️Randomization → Number assigned to each intervention.
    3️Follow-up → Dropouts and reasons for loss to follow-up.
    4️Analysis → Final number of participants analyzed per group.

  1. The Title Does Not Indicate the Study Design

Lack of Transparency in the Title

  • The title does not specify the study is an RCT, which is a major issue for clarity and indexing in scientific databases.
  • Best practices require that the title of an RCT explicitly mention its design.

Suggestions:

  • Revise the title to explicitly state the study design.

Example of a corrected title:
"Effects of Different Isometric Exercise Modalities on Myocardial Work in Trained Hypertensive Patients with Ischemic Heart Disease: A Randomized Controlled Trial."

  1. Sample Size and Power Analysis

Lack of Power Calculation

  • The manuscript is described as a pilot study, but no formal sample size calculation is provided.
  • Without a power analysis, it is unclear whether the study is adequately powered to detect meaningful differences.

Suggestion

  • Justify the sample size, even if exploratory.
  • Acknowledge the lack of power analysis as a limitation in the discussion.

Generalizability and Sex Distribution Issues

Predominantly Male Sample

  • Only three female participants were included, limiting the applicability of the findings.
  • Given known sex-related differences in cardiovascular response to exercise, this is a critical limitation.

Suggestion

  • Acknowledge this limitation explicitly in the discussion.
  • Consider whether additional recruitment is necessary for a more balanced representation.

Ethical Considerations

Potential Conflicts of Interest: None declared.
Plagiarism Detected: No evidence of plagiarism was found.
Self-Citation Analysis: The authors cite some of their prior work, but not excessively.
Other Ethical Concerns: None identified.

Final Recommendation

The manuscript presents an interesting and clinically relevant study. However, before publication, the following revisions are necessary:

- Clearly state that this is a randomized controlled trial and provide full details on the randomization process.
-Follow CONSORT guidelines and cite them explicitly in the manuscript.
-Redesign Figure 1 to align with CONSORT standards.
-Revise the title to reflect the study design.
-Provide a sample size justification and acknowledge power limitations.
-Discuss the limitations of the sex distribution and generalizability.

Comments on the Quality of English Language

English quality of the manuscript, and while it is generally understandable, it has several linguistic and stylistic issues that could impact clarity and readability. Below is an assessment of the key areas requiring improvement:

  1. Grammar and Syntax Issues

Observations:

  • The text contains grammatical errors, particularly in subject-verb agreement, prepositions, and article usage.
  • Some sentence structures are awkward, making comprehension more difficult.
  • Use of tenses is inconsistent, sometimes switching between past and present when referring to the study.

Examples of Issues & Suggested Fixes:

Original:
"Different isometric exercise modalities induce divergent acute effects on myocardial work in trained hypertensive patients with ischemic heart disease."

Revised for clarity and accuracy:
"Different isometric exercise modalities produce distinct acute effects on myocardial work in trained hypertensive patients with ischemic heart disease."

  1. Word Choice and Precision

Observations:

  • Some phrases are imprecise or overly technical, making the manuscript harder to follow.
  • Certain terms lack clarity, potentially confusing readers who are not specialists in the field.
  • Redundant phrasing reduces readability.

Examples of Issues & Suggested Fixes:

Original:
"The echocardiographic examinations were performed using a Vivid E95® cardiovascular ultrasound with a 4.0 MHz transducer for the whole duration of the study."

Revised for clarity and conciseness:
"Echocardiographic assessments were conducted throughout the study using a Vivid E95® cardiovascular ultrasound with a 4.0 MHz transducer."

Logical Flow and Cohesion

Observations:

  • Some paragraphs lack logical transitions, making the text disjointed.
  • The discussion section has repetitive statements that do not contribute new insights.
  • Certain sections do not flow smoothly, making it difficult to follow the argument.

Suggested Fixes:

  • Improve transition sentences between paragraphs to enhance readability.
  • Avoid repetitive wording—condense statements when possible.
  • Ensure that the discussion logically builds on the results without redundancy.

  1. Scientific Tone and Formality

Observations:

  • Some sentences are too conversational or lack the formal scientific tone expected in high-impact journals.
  • The use of passive voice is inconsistent—while acceptable in scientific writing, its use should be balanced for clarity.

Examples of Issues & Suggested Fixes:

Original:
"We enrolled 40 patients of both genders who were attending secondary prevention/rehabilitation programs at San Raffaele IRCCS of Rome."

Revised for a more formal tone:
"A total of 40 patients (both male and female) participating in secondary prevention and rehabilitation programs at San Raffaele IRCCS in Rome were enrolled."

Figures and Table Captions

Observations:

  • Some figure and table captions lack clarity and do not fully explain the contents.
  • The use of abbreviations is inconsistent—some are defined, while others are not.

Suggested Fixes:

  • Ensure that all abbreviations are defined upon first use and used consistently throughout.
  • Rewrite captions for figures and tables to improve clarity and make them standalone informative elements.

Final Verdict on English Quality

Current Rating: Moderate (requires revision before publication).

Recommendations:
- The article should be professionally edited for grammar, clarity, and scientific tone.
- Sentences should be shortened and structured more clearly to improve readability.
- The discussion section should be revised to avoid redundancy.
- Technical terms should be used consistently, with definitions where necessary.

Suggested Next Steps:
A professional English-language editing service or detailed revision by a native English-speaking scientific editor would significantly improve the manuscript’s quality.

Author Response

The manuscript explores the acute effects of different isometric exercise modalities on myocardial work in trained hypertensive patients with ischemic heart disease (IHD). The study is well-structured, and the research question is relevant, given the increasing interest in non-pharmacological interventions for hypertension management. The methodology is appropriate, utilizing a randomized controlled trial design with echocardiographic assessments, including speckle tracking, to evaluate myocardial work.

However, several critical aspects require clarification and improvement before the manuscript can be considered for publication. Notably:

  • The study is not explicitly labeled as a randomized controlled trial (RCT), despite having characteristics of one.

Thank you for this observation. The present study is a randomized pilot study. In the revised version of the manuscript we added this information to the title.

  • The randomization process is unclear, and there is no mention of allocation concealment or blinding procedures.

Thank you for this comment. In the revised version of the manuscript we have implemented this section

  • The study does not conform to CONSORT guidelines, which are the standard for reporting RCTs.

We have

  • Figure 1 (study flow diagram) does not align with CONSORT requirements, missing essential details such as participant recruitment, randomization method, and dropouts.

Thank you for this observation. We have changed figure 1 accordingly

  • The title does not specify the study design, which is a requirement for transparency in clinical research.

In the revised version of the manuscript we  specified that this is a  randomized pilot study in the title.

The findings suggest that handgrip exercise may be a more hemodynamically tolerable modality compared to bilateral knee extension. While this conclusion is supported by the data, additional contextual discussion regarding its clinical significance and potential implications for exercise prescription in hypertensive patients with IHD would strengthen the manuscript.

I appreciate the opportunity to review this study, which addresses an important topic in cardiovascular rehabilitation and exercise science. The authors’ efforts in conducting this research are commendable, and I hope my comments contribute to enhancing the clarity and impact of their work.

Key Strengths

Relevant Research Question: The study addresses a pertinent clinical issue regarding the safety and efficacy of isometric exercise in hypertensive patients with IHD.
Methodological Rigor: The use of echocardiographic assessments, particularly speckle tracking, is a strength, providing objective and reliable data on myocardial work.
Well-Organized Presentation: The manuscript follows a logical structure, and the results are presented with appropriate statistical analyses.

Major Concerns and Recommendations

  1. Study Design and Transparency Issues

Unclear Randomization and Lack of Explicit RCT Declaration

  • The study appears to be a randomized controlled trial (RCT), but the authors do not explicitly declare this in the manuscript.

Thank you. In the revised version of the manuscript we  specified that this is a  randomized pilot study in the title and in the methods session

  • There is no clear explanation of how the randomization was conducted. Was it computerized? Was there allocation concealment?

Thank you, In the revised version of the manuscript the randomization process has been described in the method session

  • The absence of blinding details raises concerns about potential biases in data collection and interpretation.

We are sorry for the missing information. This was a single-blinded study in which  the operator who performed the offline analysis of echocardiography variables was unaware of the patient allocation. This is now clearly stated in the method session

  • The study should explicitly declare itself as an RCT in the title, abstract, and methods section.

The nature of the study (randomized pilot study) is now explicitly declared in the title abstract and methods session.

Suggestion: Clearly state in the methods section that this is an RCT and provide details on:

  • Randomization method (e.g., computer-generated, sealed envelopes, etc.).
  • Allocation concealment (e.g., who assigned participants to groups, how was bias minimized?).

There was not allocacion concealment

  • Blinding procedures, if any.

This was a single -blinded study.  The blinding procedure has been specified in the methods session.

  1. Lack of Compliance with CONSORT Guidelines

No Mention of CONSORT, Missing Key Information

  • The manuscript does not cite the CONSORT guidelines, which are essential for reporting RCTs.
  • Essential components missing:
    • No flow diagram of participant allocation following CONSORT structure.

The required flow diagram has been added

    • No reporting of dropouts, exclusions, or adherence rates.

Thank you. These data have been added

    • No explicit statement of primary and secondary outcomes in line with CONSORT recommendations.

In the revised version of the manuscript  we have clearly stated that the study complies with  CONSORT guidelines

Suggestion:

  • The authors must cite CONSORT guidelines and restructure their methodology and results to align with them.

In the revised version of the manuscript  we have clearly stated that the study complies with  CONSORT guidelines. The methodology has been improved accordingly

  • Include a proper CONSORT-compliant flow diagram (see next section).

We have now changed figure 1  and we have included a CONSORT-compliant flow diagram 

  • Add transparency regarding participant losses, exclusions, and adherence to the intervention.

 Thank you. These informations have been added

  1. Figure 1 (Study Flow Diagram) is Incomplete and Non-CONSORT Compliant

Critical Missing Information:

  • The current Figure 1 does not provide the required details for a clinical trial.
  • Missing elements:
    • Total number of participants screened, excluded, and randomized.

Thank you. This information has been added

  • Justification for exclusions before randomization.

               Thank you. This information has been added

    • Breakdown of how many participants completed each stage of the study.

This is not applicable since the study consisted in a single session of exercise

    • Dropouts and reasons for attrition in each group.

Thank you for this comment. In the result paragraph it is now clearly stated that there were no dropouts and all included patients completed the study and were analysed.

Suggestion

  • Redesign Figure 1 to follow CONSORT-compliant flow diagram standards, clearly showing:
    1️Enrollment → Number of patients assessed for eligibility.

Thank you. This information has been added

               2️Randomization → Number assigned to each intervention.
                Thank you. This information has been added (figure 1)

3️Follow-up → Dropouts and reasons for loss to follow-up.
In the present study there were no dropouts or patients lost to follow-up

              4️Analysis → Final number of participants analyzed per group.

                Thank you. This information has been added (figure 1)

  1. The Title Does Not Indicate the Study Design

Lack of Transparency in the Title

  • The title does not specify the study is an RCT, which is a major issue for clarity and indexing in scientific databases

In the revised version of the manuscript the tiottle has been changed and now it includes the following sentence: a randomized pilot study.

  • Best practices require that the title of an RCT explicitly mention its design.

Suggestions:

  • Revise the title to explicitly state the study design.

Example of a corrected title:
"Effects of Different Isometric Exercise Modalities on Myocardial Work in Trained Hypertensive Patients with Ischemic Heart Disease: A Randomized Controlled Trial."

 Tank you for your suggestion; we changed the title as follow:

Effect of different isometric exercise modalities  on myocardial work in trained hypertensive patients with ischemic heart disease: a randomized pilot study

  1. Sample Size and Power Analysis

Lack of Power Calculation

  • The manuscript is described as a pilot study, but no formal sample size calculation is provided.
  • Without a power analysis, it is unclear whether the study is adequately powered to detect meaningful differences.

We understand this comment and we have underlined this point in the  limitation paragraph:

Since this research was conceived as a pilot study, it included a small sample size and lacked adequate power analysis. These considerations make unclear whether the study was adequately powered to detect meaningful differences. Therefore, caution is needed in drawing general conclusion by data provided by this study.

Suggestion

  • Justify the sample size, even if exploratory.
  • Acknowledge the lack of power analysis as a limitation in the discussion.

Thank you in the revised version of the manuscript we specified better this point in the limitation paragraph, as specified above.

Generalizability and Sex Distribution Issues

Predominantly Male Sample

  • Only three female participants were included, limiting the applicability of the findings.

Given known sex-related differences in cardiovascular response to exercise, this is a critical limitation.

Thank you for this observation. We have think that the limitation paragraph already contains the following sentence:

  • The study enrolled only 3 female patients in the active arms. Due to this unbalanced gender distribution of our sample, the results of this research cannot be extended to women. Further studies focusing on the effects of  isometric exercise in female with hypertension and IHD will therefore be necessary"

Suggestion

  • Acknowledge this limitation explicitly in the discussion.

Done

  • Consider whether additional recruitment is necessary for a more balanced representation.

In the revised version of the manuscript, in the limitation paragraph, we have underlined the necessity  to conceive new studies focusing on the effects of  isometric exercise in female with hypertension and IHD will therefore be necessary

Ethical Considerations

Potential Conflicts of Interest: None declared.
Plagiarism Detected: No evidence of plagiarism was found.
Self-Citation Analysis: The authors cite some of their prior work, but not excessively.
Other Ethical Concerns: None identified.

Final Recommendation

The manuscript presents an interesting and clinically relevant study. However, before publication, the following revisions are necessary:

- Clearly state that this is a randomized controlled trial and provide full details on the randomization process.
-Follow CONSORT guidelines and cite them explicitly in the manuscript.

-Redesign Figure 1 to align with CONSORT standards.

Done
-Revise the title to reflect the study design.

Done
-Provide a sample size justification and acknowledge power limitations.

-Discuss the limitations of the sex distribution and generalizability.

We have already the following sentence:

Comments on the Quality of English Language

English quality of the manuscript, and while it is generally understandable, it has several linguistic and stylistic issues that could impact clarity and readability. Below is an assessment of the key areas requiring improvement:

  1. Grammar and Syntax Issues

Observations:

  • The text contains grammatical errors, particularly in subject-verb agreement, prepositions, and article usage.
  • Some sentence structures are awkward, making comprehension more difficult.

Use of tenses is inconsistent, sometimes switching between past and present when referring to the study.  

Thank you for your observations. Grammar and syntax have been improved thanks to the English language service of MDPI

  •  

Examples of Issues & Suggested Fixes:

Original:
"Different isometric exercise modalities induce divergent acute effects on myocardial work in trained hypertensive patients with ischemic heart disease."

Revised for clarity and accuracy:
"Different isometric exercise modalities produce distinct acute effects on myocardial work in trained hypertensive patients with ischemic heart disease."

  1. Word Choice and Precision

Observations:

  • Some phrases are imprecise or overly technical, making the manuscript harder to follow.
  • Certain terms lack clarity, potentially confusing readers who are not specialists in the field.
  • Redundant phrasing reduces readability.

Word choice and precision have been improved thanks to the English language service of MDPI

Examples of Issues & Suggested Fixes:

Original:
"The echocardiographic examinations were performed using a Vivid E95® cardiovascular ultrasound with a 4.0 MHz transducer for the whole duration of the study."

Revised for clarity and conciseness:
"Echocardiographic assessments were conducted throughout the study using a Vivid E95® cardiovascular ultrasound with a 4.0 MHz transducer."

 Thank you we adopted your suggestion

Logical Flow and Cohesion

Observations:

  • Some paragraphs lack logical transitions, making the text disjointed.
  • The discussion section has repetitive statements that do not contribute new insights.
  • Certain sections do not flow smoothly, making it difficult to follow the argument.

Suggested Fixes:

  • Improve transition sentences between paragraphs to enhance readability.
  • Avoid repetitive wording—condense statements when possible.
  • Ensure that the discussion logically builds on the results without redundancy.

  1. Scientific Tone and Formality

Observations:

  • Some sentences are too conversational or lack the formal scientific tone expected in high-impact journals.
  • The use of passive voice is inconsistent—while acceptable in scientific writing, its use should be balanced for clarity.

Examples of Issues & Suggested Fixes:

Original:
"We enrolled 40 patients of both genders who were attending secondary prevention/rehabilitation programs at San Raffaele IRCCS of Rome."

Revised for a more formal tone:
"A total of 40 patients (both male and female) participating in secondary prevention and rehabilitation programs at San Raffaele IRCCS in Rome were enrolled."

  Thank you we adopted your suggestion

Figures and Table Captions

Observations:

  • Some figure and table captions lack clarity and do not fully explain the contents.

Thank you for this comment. We have implemented captions of figure 2,3, 4 and 5. The caption of figure 5 was wrong and we corrected it.

  • The use of abbreviations is inconsistent—some are defined, while others are not.

We have checked and improved the use of abbreviations

Suggested Fixes:

  • Ensure that all abbreviations are defined upon first use and used consistently throughout.

done

  • Rewrite captions for figures and tables to improve clarity and make them standalone informative elements.

              done

Final Verdict on English Quality

Current Rating: Moderate (requires revision before publication).

Recommendations:
- The article should be professionally edited for grammar, clarity, and scientific tone.

Thank you;  the manuscript has been revised  by the English-language editing service of MDPI

- Sentences should be shortened and structured more clearly to improve readability.

Thank you the structure of sentences has  been revised  by the English-language editing service 

- The discussion section should be revised to avoid redundancy.
Technical terms should be used consistently, with definitions where necessary.

Suggested Next Steps:
A professional English-language editing service or detailed revision by a native English-speaking scientific editor would significantly improve the manuscript’s quality.

Thank you the manuscript has been revised  by the English-language editing service of MDPI

Reviewer 2 Report

Comments and Suggestions for Authors

Greetings to the authors, after reading your manuscript my comments are as follows. 

In the introduction the authors briefly  mention in lines 66 to 70 speckle tracking echocardiography. The authors should expand this discussion and also mention that there are a number of factors that can have an impact on the parameters of speckle tracking such as dilate cardiomyopathy, heart failure and diabetes they may refer to DOI: 10.3390/diagnostics12010035 and doi: 10.3390/jcm13144037.
These should be mentioned as a significant number of the enrolled patients have heart failure and several also have diabetes.

regarding the study population, while the authors have enrolled a significant number of patients, they do mention that the patients have had previous ischemic events without having current ongoing ischemia. However I am surprised that in table 1 when illustrating the ejection fractions of the patients, the lowest is moderately reduced with most patients having near normal EF. Was this also a criteria for the inclusion in the study ? since this is a very important aspect as it seems that all these patients have been successfully revascularized.

The authors mention in the discussion the complexity of the pharmacological treatment undergone by the enrolled patients. Perhaps the authors should have mentioned in the manuscript which patients were undergoing which type of medication as given the complexity of thei diseases most likely many of them were under treatment with ace inhibitors or ARNI. This is worth mentioning since one of the main focuses of the manuscript revolves around BP.

Author Response

In the introduction the authors briefly  mention in lines 66 to 70 speckle tracking echocardiography. The authors should expand this discussion and also mention that there are a number of factors that can have an impact on the parameters of speckle tracking such as dilate cardiomyopathy, heart failure and diabetes they may refer to DOI: 10.3390/diagnostics12010035 and doi: 10.3390/jcm13144037. These should be mentioned as a significant number of the enrolled patients have heart failure and several also have diabetes.

Thank you for this comment We added a short sentence in the introduction and expanded this concept in the limitation paragraph of the discussion.

In this study LV hemodynamic changes were assesses noninvasively through speckle-tracking echocardiography and, in particular, myocardial work. Although myocardial work is believed to provide a more accurate assessment of the LV function than LVEF and LVGLS, it has several technical limitations: these include poor image quality that does not allow correct endocardial border delineation and partial load-dependency [38]. Overall, these limitations could prevent the extension of the results of the present study. Finally, the use of myocardial work to investigate the cardiac response to exercise is so far very limited and further research are needed to establish its utility in this field.  

Regarding the suggested references, we thank the reviewer for these suggestions, however we do not think that they are relevant for this study

regarding the study population, while the authors have enrolled a significant number of patients, they domention that the patients have had previous ischemic events without having current ongoing ischemia. However I am surprised that in table 1 when illustrating the ejection fractions of the patients, the lowest is moderately reduced with most patients having near normal EF. Was this also a criteria for the inclusion in the study ? since this is a very important aspect as it seems that all these patients have been successfully revascularized.

Thank you for this comment. We enrolled patients who had no clinical history of heart failure irrespectively of their ejection fraction. Patients with incomplete revascularization were excluded from the study. In the revised version of the  manuscript we clarified this point in the methods session

The authors mention in the discussion the complexity of the pharmacological treatment undergone by the enrolled patients. Perhaps the authors should have mentioned in the manuscript which patients were undergoing which type of medication as given the complexity of thei diseases most likely many of them were under treatment with ace inhibitors or ARNI. This is worth mentioning since one of the main focuses of the

manuscript revolves around BP.

Thank you for this observation. No patient was taking ARNI, as none of them had been diagnosed with heart failure

There were no differences in pharmacological therapy between the three groups.  In the revised version of the manuscript we have now specified this point  

Round 2

Reviewer 1 Report

Comments and Suggestions for Authors

Dear Authors,

I appreciate the opportunity to review your revised manuscript titled "Effect of Different Isometric Exercise Modalities on Myocardial Work in Trained Hypertensive Patients with Ischemic Heart Disease: A Randomized Pilot Study." I commend you for your thorough and thoughtful revisions, which have significantly improved the clarity, transparency, and overall quality of the study.

You have effectively addressed the key concerns raised in the previous review, particularly by:

  • Clearly stating the study design as a randomized pilot study in the title, abstract, and methods section.
  • Providing further details on the randomization process, including the use of a computer-generated sequence with permuted blocks.
  • Clarifying the single-blind design, specifying that the echocardiographic analysis was performed by an operator unaware of patient allocation.
  • Ensuring compliance with CONSORT guidelines, including revising the study flowchart (Figure 1) to provide details on participant screening, exclusions, and randomization.
  • Revising the title to explicitly indicate the study design.
  • Acknowledging the lack of a formal power analysis in the limitations section.
  • Discussing the gender imbalance in the study population and emphasizing the need for future research in female participants.
  • Submitting the manuscript to professional English language editing, improving clarity, coherence, and scientific rigor.
  • Revising figure and table captions for greater clarity and consistency.

The only remaining limitation is the lack of allocation concealment, which has been duly acknowledged in the manuscript.

Overall, your revisions have strengthened the manuscript considerably, and I believe it now provides a clearer and more robust contribution to the literature on isometric exercise and cardiovascular health. I appreciate your efforts in addressing the reviewers’ comments comprehensively.

Best regards,

Author Response

The only remaining limitation is the lack of allocation concealment, which has been duly acknowledged in the manuscript.

We would like to thank the reviewer for his commitment and important contribution
In the new version of the manuscript we reported the lack of allocation concealment among the limitations of this study

Reviewer 2 Report

Comments and Suggestions for Authors

“Thank you for this comment We added a short sentence in the introduction and expanded this concept in the limitation paragraph of the discussion.”

The introduction has been expanded quite slightly from my point of view, the suggested expansion did not involve cardiac work rather the imaging process and the factors that may effect GLS, as suggested.

“Thank you for this comment. We enrolled patients who had no clinical history of heart failure irrespectively of their ejection fraction. Patients with incomplete revascularization were excluded from the study. In the revised version of the manuscript we clarified this point in the methods session”

The authors have replied that there was no history of heart failure in the study patient slot, however, these are coronary patients which had also had acute coronary events, and this is highly unlikely. If that was the case the authors should have included only chronic coronary patients or patients with coronary artery disease that have not had acute coronary events. Otherwise, the statement that these patients do not have any degree of heart failure cannot apply to this study lot in particular, at least when referring to systolic heart failure, without even taking into equation diastolic heart failure. When including ejection fraction in the equation as was the case of the current study, the authors cannot simply mention that they did not take EF into consideration.

“Thank you for this observation. No patient was taking ARNI, as none of them had been diagnosed with heart failureThere were no differences in pharmacological therapy between the three groups. In the revised version of the manuscript we have now specified this point”

From my point of view, this is a superficial response on behalf of the authors. The patients included in study were taking the exact same antihypertensive medication in the exact same dosage ? not only that, but considering patients with acute coronary events were enrolled none of them were on ARNI ? How can not one of the included patients be diagnosed with any degree of heart failure ? The authors' reply in this regard raises several concerns.

Author Response

The introduction has been expanded quite slightly from my point of view, the suggested expansion did not
involve cardiac work rather the imaging process and the factors that may effect GLS, as suggested.

Thank you for this comment. We have now expanded the introduction paragraph to highlight the limitations
and potential applications of speckle tracking echocardiography. In the added text, we felt it appropriate to
add new references, including those previously suggested by the reviewer:

“This ultrasound technique, through the measurement of strain and strain rate, assess the percentage of
deformation of the myocardium and allows to perform a comprehensive evaluation of LV and left atrial (LA)
function [9-11]. Despite several factors including image quality, modest reproducibility of measurements and
cardiac rhythm, can affect the accuracy and reliability of strain-dependent measurements such as LV global
longitudinal strain (LVGLS) [12], speckle tracking echocardiography has demonstrated significant value for
patient management and in guiding towards optimum treatment strategies, particularly in patients with
advanced cardiac diseases [13,14].”

The authors have replied that there was no history of heart failure in the study patient slot, however, these
are coronary patients which had also had acute coronary events, and this is highly unlikely. If that was the
case the authors should have included only chronic coronary patients or patients with coronary artery
disease that have not had acute coronary events. Otherwise, the statement that these patients do not have
any degree of heart failure cannot apply to this study lot in particular, at least when referring to systolic heart
failure, without even taking into equation diastolic heart failure.
When including ejection fraction in the equation as was the case of the current study, the authors cannot
simply mention that they did not take EF into consideration.

We appreciated this comment of the reviewer.
During the screening visit, we adopted a clinical criteria according to the ESC guidelines that define heart
failure as a clinical syndrome. We simply ruled out patients who had experienced in one or more occasions
(or who had during the screening visit) symptoms and/or signs of heart failure. Clearly, we are aware that
patients of the present study are at high risk of developing heart failure due to the heart damage suffered in
the past. Moreover, we admit that a proportion of the population had asymptomatic mild left ventricular
systolic dysfunction. However, at the time of recruitment, none of them had signs or symptoms (or had had
in the past) of heart failure. Regarding the diastolic function, indirect metrics of left ventricular filling
pressure were well below the upper normal limits.
In the revised version of the manuscript we added the distribution of the population according to LVEF.
Moreover we added a new comment in the discussion paragraph (see below)

From my point of view, this is a superficial response on behalf of the authors. The patients included in study
were taking the exact same antihypertensive medication in the exact same dosage ?

We apologize for our previous response but maybe we did not fully understand the concept expressed by the
reviewer. Despite there were not statistical differences between the three groups regarding cardiovascular
drugs, it is possible that different drugs may have affected the hemodynamic response to IE in a different
way. This is a very interesting point that need further investigations. However, we believe that this study, due
to its design and the small sample size, is not suitable to answer the question posed by the reviewer

“Despite there were not statistical differences in pharmacological therapy between the three groups, it is
possible that different drugs may have affected the hemodynamic response to IE in a different way. However,
this study, due to its design and the small sample size, is not suitable to provide this type of information and
this point needs to be clarified in further investigations”.

not only that, but considering patients with acute coronary events were enrolled none of them were on ARNI
?

Regarding the therapy with ARNI, we do not find it surprising that none of the patients were taking it: to date,
there is no specific recommendation for ARNI in IHD patients with asymptomatic mildly reduced or normal
ejection fraction. Three patients had EF below 40% and could have taken sacubitril-valsartan, but this was
not the case at the moment of their recruitment.

How can not one of the included patients be diagnosed with any degree of heart failure ?

We admit that our patients, having a chronic left ventricular damage and in certain cases mild left ventricular
dysfunction, had some features of heart failure and were at high risk of developing the fully heart failure
syndrome. However, at the time of recruitment, none of them had signs and/or symptoms (or had had in the
past) of heart failure. We considered them having ischemic heart disease without heart failure.
In the revised version of the manuscript we specified this point in the discussion/limitation paragraph:

“It is worth mentioning that patients enrolled in the present study, having a chronic LV damage and, in
several cases mild LV dysfunction, had some features of heart failure and were at high risk of developing the
fully heart failure syndrome. However, at the time of recruitment, none of them had signs or symptoms (or
had had in the past) of heart failure and were not taking the full drug treatment specifically used for heart
failure. We considered them having IHD without heart failure. It is possible, however, that the response of
myocardial work to IE was different across EF values. In this study we did not consider this point due to the
small sample size and it should be investigated in further research”

Round 3

Reviewer 2 Report

Comments and Suggestions for Authors

The manuscript has been improved over this round of revisions and the authors have explained the questionable elements within the text. Overall I have no more additional comments, the decision belongs to the editors after this round of revisions.